# Brain Signaling of Indispensable Amino Acid Deficiency

**DOI:** 10.3390/jcm11010191

**Published:** 2021-12-30

**Authors:** Dorothy W. Gietzen

**Affiliations:** Department of Anatomy, Physiology and Cell Biology, University of California, Davis, CA 95616, USA; dwgietzen@ucdavis.edu

**Keywords:** protein synthesis initiation, neural signaling, anterior piriform cortex, transfer RNA, GCN2

## Abstract

Our health requires continual protein synthesis for maintaining and repairing tissues. For protein synthesis to function, all the essential (indispensable) amino acids (IAAs) must be available in the diet, along with those AAs that the cells can synthesize (the dispensable amino acids). Here we review studies that have shown the location of the detector for IAA deficiency in the brain, specifically for recognition of IAA deficient diets (IAAD diets) in the anterior piriform cortex (APC), with subsequent responses in downstream brain areas. The APC is highly excitable, which makes is uniquely suited to serve as an alarm for reductions in IAAs. With a balanced diet, these neurons are kept from over-excitation by GABAergic inhibitory neurons. Because several transporters and receptors on the GABAergic neurons have rapid turnover times, they rely on intact protein synthesis to function. When an IAA is missing, its unique tRNA cannot be charged. This activates the enzyme General Control Nonderepressible 2 (GCN2) that is important in the initiation phase of protein synthesis. Without the inhibitory control supplied by GABAergic neurons, excitation in the circuitry is free to signal an urgent alarm. Studies in rodents have shown rapid recognition of IAA deficiency by quick rejection of the IAAD diet.

## 1. Introduction and History

Of the 20 amino acids that are incorporated into the body’s protein, the 9 essential, i.e., indispensable amino acids (IAAs) are those that are neither stored nor synthesized in the body in adequate amounts to support the needs of protein synthesis and metabolism (Table 1). In particular, the limitation of IAAs will lead to dysfunction within the brain, including seizures [1]. The IAAs include the branched-chain AAs, leucine, isoleucine and valine, plus the sulfur-containing AAs, methionine (which can be partially replaced by cysteine), tryptophan, phenylalanine, lysine, threonine and histidine. This distinction between dispensable AAs and IAAs dates to the early metazoans, reviewed in [2]. In contrast to plants and organisms that can synthesize all their AAs internally, omnivores and herbivores depend on their diets to supply the IAAs that are essential for maintaining the synthesis of proteins, hormones and other body components, as well as for structural integrity, thus the term dietary IAAs. These must be acquired, most often from food, so the abilities to recognize and reject an IAA deficiency in the current diet, to seek better food and to recognize the repletion quickly, are crucial for health, healing, reproduction, and indeed, survival.

While nitrogen containing foods have been known to be important parts of the human diet for over 200 years [3], the earliest observations showing growth failure from IAA deficient (IAAD) diets were made in the early 1900s [4]. Additionally, early in the 1900s, it was reported that animals did not grow normally when they were fed diets based on gelatin, which is devoid of the IAA, tryptophan, as well as deficient in several other IAAs. Thus, gelatin was shown not to be an adequate source of protein for the support of growth, reviewed in [5].

McCoy and colleagues [6], in completing the list of IAAs, reported the isolation of the final IAA, threonine. So, a mixture of L-AAs including all the IAAs could substitute for adequate protein in the diet. Then, each, or any group, of the IAAs could be eliminated to study their effects with precision. Later, it was found that animals rejected several types of imbalanced or devoid IAAD diets. The work to answer the many questions raised by these discoveries up to 1970 was thoroughly reviewed by Harper et al. [7]. By reporting that decreased food intake occurs *before* growth retardation or other deleterious effects, Harper and Rogers [8] showed that altered eating behavior was the first sign of sensing IAAD. Then, because growth can be maintained in animals on an imbalanced IAAD diet if intake was maintained by tube feeding or providing the animals no other food choice, reduced food intake was recognized to be the cause of, rather than resulting from, the poor growth reported in several animal species [7]. Indeed, a reduction in feeding was observed as early as 28 min after introduction of a threonine devoid diet, a finding correlated with the changes in concentration of threonine in the plasma (Figure 1), reviewed in [9], and later supported by Gloaguen et al. [10].

This early response is likely due to the efficiency of the AA transporters with overlapping affinity at the blood–brain barrier [11,12].

## 2. The Search for the Sensor

Was the sensor in the brain, any of the known sensory pathways, or in the periphery? Because occluding the nares, so that the animals could not smell, did not affect their ability to recognize the IAAD, clearly, smell was not the sensory mechanism [13]. Similarly, taste was not the initial mechanism [14] nor was palatability of the diet [15]. In addition, the recognition of IAAD also does not require intact liver [16], vagal [17], gastric [18], or adrenal [19] function.

The importance of the brain in recognizing and signaling for an animal to reject its diet was first reported by Quinton Rogers and Philip Leung. Infusions of the limiting IAA into the carotid artery restored eating of the IAAD diet, but similar infusions into the jugular vein required 40-fold higher concentrations for the same effect (Figure 2) [20], a finding replicated in cockerels [21].

The first results showing that the responses to IAAD diets are in the brain were seen when Leung and Rogers [20] cannulated rats in either the jugular vein (JV) or carotid artery (CA) and infused threonine or saline into each cannula. Food intakes of the threonine imbalanced diet are shown for the first 4 days after onset of the infusions. Clearly, threonine in the CA blocked the feeding depression to the IAAD diet, while the same infusion into the JV did not.

Tews et al. [22,23,24] reported the mechanism for getting IAA imbalance in the brain via competition for AA transporters [25] but did not localize the effect within the brain. Nonetheless, clearly the brain housed the sensor that responded to the loss of the limiting IAA.

The next question was: where in the brain is the sensor? To begin the studies locating the sensor for IAAD, Rogers and Leung made electrolytic lesions in thirteen brain areas, plus the pituitary and olfactory bulbs. While investigating various brain areas that could house the sensor, a student misplaced the surgical apparatus and lesioned an area that had not been under consideration, as it was further anterior than any areas studied previously. Surprisingly, the animals with these far-anterior lesions were unable to recognize the IAAD diet and continued eating it, whereas lesions in the hypothalamus and other brain areas known to be involved in the control of food intake had no such effect Table 2 [26].

Evaluation of brain slices bearing the lesions that had resulted in the rats’ inability to recognize an IAAD showed that the area sensitive to IAAD diets was in the anterior-lateral cortex, specifically, the Anterior Piriform Cortex (APC) [27]. This brain area is the most primitive of all animal cortices and also serves as the primary olfactory cortex [28]. These lesion data showed that the limbic–rhinencephalic areas associated with olfactory information, including the APC and the amygdala, were important in the initial decreased feeding responses in rats fed an IAA imbalanced or devoid diet [27,29,30]. However, the concentration of the limiting IAA was not altered in the amygdala or in any of the three areas of the hypothalamus that are involved in feeding. This illustrates the importance of IAA concentrations in the brain areas implicated in IAAs. So, neither the amygdala nor the feeding centers in the hypothalamus could serve as the primary sensor; by default, the APC was accepted as the brain’s sensory area for recognizing an IAAD diet. In addition, the APC is in or near a highly excitable area determined by Gale and colleagues, who injected progressively smaller doses of a seizure-inducing drug into various discrete brain areas to determine the most excitable area, which they termed the Area Tempestas [31]. Taken together, the evidence was consistent with a role for the APC as the prime candidate for rapid activation by disturbances of IAA balance.

## 3. Methods: Importance of the Prefeeding Method in Studying the Earliest Response

To study the time course of IAAD recognition, the subjects must be prepared properly. Prefeeding diets that do not overload the body with protein must be used, as an IAAD diet will not be noticed when the plasma is replete with an excess of IAAs. As an example of pretrial treatments, in Beverly’s studies [32,33,34,35], preparation of the subjects for rapid recognition of an IAAD diet, included the following: Rats were fed a low protein basal diet that was limiting in the IAA to be tested for at least a week. On this diet, they grew as well as the controls, but this prefeeding protocol aided the testing for sensing a deficiency because the diet contained minimal amounts of protein and particularly, was limited (but not devoid) of the IAA to be studied. In such animals, the test IAAD diet, either imbalanced or devoid of a single IAA, depletes the limiting IAA rapidly in the plasma and brain. Amino acids in our studies were determined by chromatography using a Beckman 7300 amino acid analyzer (Palo Alto), newer models will continue to be available. With this feeding protocol, Koehnle [36] recorded the decreasing weights of food cups, using on-line computerized balances, in intervals of a few seconds. Analysis of the meal patterns and durations revealed that 50% of the animals in the study had stopped eating an IAAD meal in 20 min, and the time to the second meal was significantly delayed. So, the mechanism underlying the sensing and rejection of IAAD was rapidly deployed. Therefore, to investigate the earliest sensory mechanism for IAAD, the subjects need to be studied within the first ½ h after introducing the IAAD diet

Many years were devoted to determining the mechanism of the IAAD sensor and the neurochemistry involved. It is important that the sensing and rejection of an IAAD must be rapid, as signals due to the missing IAA will soon stimulate protein degradation [37] resulting in functional losses, in the brain in particular. Beverly et al. [32,33,34,38] reported a series of results using injections into the APC that became the basis for the subsequent discovery of the role of protein synthesis in the activation of the APC after IAAD diet ingestion. These experiments showed the specificity of the response to the limiting IAA. After injections of threonine into the APC, the feeding depression to a threonine (but not an isoleucine) imbalanced diet was blocked. Conversely, injections of threonine had no effect on the depressed feeding of an isoleucine imbalanced diet.

It is important to note that this effect could be blocked by the protein synthesis inhibitor, puromycin, injected with the vehicle or IAA into the APC using the above method [33]. The time course of the effective injections suggested that diffusion or metabolism might account for these observations. Diffusion of L-[^14^C]-threonine was rapid (<15 min) but limited, as most of the label was recovered from within 1 mm of the injection site [34]. Because decreases of an IAA had been linked to protein synthesis in single-cell systems [39], puromycin, a protein synthesis blocker, was added to the injection protocol. These additions blocked the amino acid restoration of intake of the IAAD diet but did not alter intake of the basal control diet. Similarly, puromycin blocked the usual choice for control over IAAD diets [33]. These findings led to the suggestions that (1) a reduced concentration of the limiting IAA in the APC, along with (2) changes in protein synthesis, are associated with the mechanisms involved in recognizing an IAAD diet. Thus, we were able to state that “inability to make a crucial protein in the APC could lead to a neural signal that protein synthesis is at risk” [26].

The mechanisms for this rapid recognition of IAAD involve an enzyme important in the initiation of new protein synthesis: General Control Nonderepressible 2 (GCN2) and its target, phosphorylated eIF2α can be seen in Figure 6 of [40,41] and has been the subject of many reviews. For example, in 2021 alone, see [42,43,44,45]. The mechanisms involve an accumulation of uncharged tRNA because the limiting IAA that is deficient in the plasma and brain cannot adequately charge its cognate tRNA. Therefore, the concentration of charged tRNA is reduced to such a low level that the initiation of new protein synthesis cannot proceed [46]. Furthermore, nanomolar injections of L-IAA alcohols, which inhibit tRNA acylation by competition with their cognate IAA, replicate a deficiency of IAAs showing both the rejection of diet and other similar biochemical responses [40], supporting these findings.

The stereospecific tRNA inhibitor blocked intake of the the basal diet at 40 min after the onset of feeding. The results shown in Figure 3 show the importance of tRNA charging, which is required for the initiation of protein synthesis [40]. GCN2 is also involved in the control of a fibroblast growth factor (FGF21) via the activating transcription factor 4 (ATF4) pathway [47]. GCN2 is now recognized as the primary sensor for IAA deficiency, as recently reviewed [48].

**Figure 3 jcm-11-00191-f003:**
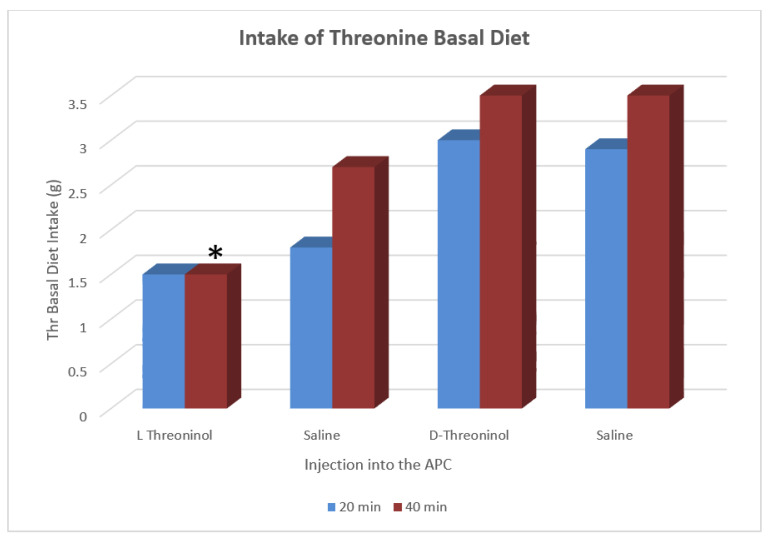
The mechanism of recognizing IAAD in the APC is shown by injections, as in Figure 4, of the tRNA analogue threoninol, which competes for threonine in charging its cognate tRNA. The results using GCN2 knockout mice also showed that the enzyme GCN2 is required for recognition of an IAAD diet [40]. Star ***** indicates significance of L-threoninol injections into the APC on intake of the threonine basal diet.

The output neurons of the APC, the pyramidal cells of layer II, are glutamatergic excitatory cells. They are under the inhibitory control of GABAergic proteins in the same area [49]. These proteins are important in controlling the output circuitry of the APC, but they have a very short half-life [50], so in the absence of new protein synthesis the inhibitory elements are lost from the neural membrane within minutes [50]. As a result, these rapidly turning-over proteins, including GABAergic inhibitory receptors and ion transporters, cannot be replaced, and the APC cannot retain the normal balance between the stimulatory and inhibitory neurons in the circuitry. Given this disinhibition, the glutamatergic excitatory neurons in the APC are free to send excitatory signals to a wide variety of brain areas involved in motor activity and food intake [51], resulting in rejection of the deficient diet and increased activity. Tract tracing studies using a stain for axonal projections show brain areas with direct axonal projections from the APC [51], including to the lateral hypothalamus (Figure 5).

Consistent with these observations, Monda et al. showed that L-threonine injections into the APC increased food intake and neuronal activity in the lateral hypothalamus, and sympathetic nervous system activity was reduced in rats that were fed a threonine devoid diet [52]. In view of this report, Blevins and colleagues looked at the effects of threonine injections in the lateral hypothalamus on the intake of IAAD or corrected diets, specifically a threonine devoid diet or a threonine corrected diet. After these injections, intake of the devoid diet was increased during the first 6 h, while intake of the corrected diet was not affected. These studies showed that the APC and lateral hypothalamus both take part in the circuitry affected by IAAD diets [53]. Because the output cells of the APC are glutamatergic, and their axons project directly to the lateral hypothalamus, reviewed in [51], one would expect that glutamate’s N-methyl D-aspartate (NMDA) receptors in the lateral hypothalamus may be involved in this circuitry. To study this issue, Blevins et al. used the NMDA antagonist, D-2-Amino-5-phosphopentanoic acid (D-AP5), showing that this antagonist, when injected into the APC, increased intake of the IAAD diet, but injected into the lateral hypothalamus, decreased IAA devoid diet preference vs. the corrected diet, while it did not affect intake of the devoid diet. The glutamate antagonist NBQX (FG9202), when injected into the APC increased IAAD diet intake, NMDA injection into the lateral hypothalamus increased intake of the AA corrected diet but did not change IAA devoid diet intake. These results show that glutamate from the APC’s axons acts in the lateral hypothalamus to affect IAAD diet intake [54].

Initial behavioral responses includes, along with rejection of the IAAD diet, an increased motor activity that is seen as the animals digging into the food cup and searching their spaces for a different food source. Of additional interest are the subsequent behaviors, such as increased activity, foraging, sampling of any available new foods, and the like, along with learned aversions.

## 4. Conditioned Taste Aversion

It is well known that, after the initial recognition of an IAAD diet in the APC, the subjects show a long-lasting rejection for the diet and or cues related to it, developing a conditioned taste aversion (CTA) [15,55]. CTA is found throughout evolution, as reported in snails [56], Limax, a terrestrial slug [57] and the like. The CTA can develop only after recognition of some negative effect [58], often a negative visceral response [59,60,61]. Consistent with this, involvement of the vagus nerve was shown by Dixon et al. [62]. This phenomenon has undergone extensive study and review, starting with the work of Garcia et al. [60] that demonstrated the preference for saccharin solutions shown by rats, but after pairing these solutions with gamma radiation, the animals became averse to saccharin; this is now known as the Garcia effect [60,63,64].

Other brain areas, besides the APC, are also activated within hours after the animals have eaten an IAAD diet, suggesting that they may be involved in the development of a CTA to the deficient diet. The Fos protein, expressed by the immediate-early-gene *c-fos* was increased, indicating brain activation after the initial recognition period, in brain areas including the endopiriform cortex, dorsomedial nucleus of the hypothalamus, and central nucleus of the amygdala, a pattern of labeling that was not replicated in animals fed the control diets [65]. In particular, the amygdala has been linked to CTA and was shown to affect intake of an IAAD diet [66,67,68]. In addition, large electrolytic lesions that involve multiple nuclei in the amygdala blunted the usual CTA to lithium chloride. Glutamate, injected into the basolateral amygdala (BLA) enhanced CTA, while blockade of glutamate receptors reduced CTA learning [56]. Lesions of the medial nucleus also replicated the earlier work on such aversions [66]. The central nucleus of the amygdala (CEA) contains at least 3 distinct populations of neurons, each of which inhibits the others. Increased feeding is associated with cells containing the serotonin 2A receptors, whereas decreased feeding neurons contain the kinase, PKC-delta; both are found in neurons that are considered late firing cells [69]. In addition, the insular cortex has been implicated in CTA after a few hours. Here, muscarinic receptor antagonists blocked the CTA. The synaptic connections between the BLA and the insular cortex were described by Haley et al. [70], and cFos expression has been seen by several observers in the BLA and insular cortex after hours of CTA exposure, reviewed in [56]. Further interactions between these brain areas includes the observation that brain-derived neurotrophic factor (BDNF) increased signaling capabilities of the insular cortex, and changed the duration of the CTA [71]. Directly related to this observation, conditioned preferences for the limiting IAA, or cues associated with its repletion, have been shown by several laboratories [14].

## 5. Sensing Repletion

A complete dietary supply of IAAs will increase protein synthesis and prevent proteolysis [72]. Clearly, the responses of animals to the corrected or basal control diets in the many studies cited here show increased ingestion of the balanced diet in approximately 30 min. This provides direct support for a signaling mechanism aimed at maintaining a balanced IAA profile in the diet. Signals sensing adequate levels of IAAs also stimulate anabolic and storage pathways. The IAAs, such as leucine, have been shown to activate the protein synthetic-signaling pathway associated with the mechanistic target of rapamycin complex 1 (mTORC1) that acts via stimulation of ribosomal protein S6 kinase (S6K1) and inhibition of eukaryotic translation initiation factor 4E binding protein-1 (4EBP1) [73]. The mechanistic target of rapamycin (mTOR) is also linked to the phosphoinositide 3-kinase (PI3K) pathway and interacts with other components to form mTORC1 or mTORC2 complexes [74]. The regulation of mTOR occurs via AA transport, which includes a role for glutamine, a dispensable AA [75]; the selective system A transporter substrate, alpha-Aminobutyric acid, is highly affected by glutamine in APC neurons [76]. Although we were able to rule out roles for mTORC1 and mTORC2 in detecting IAA deficiency in the APC [75], their involvement in sensing repletion in the APC may be important. In contrast to IAAD, mTOR responds to AA abundance [48]. In animals, AA supply activated mTORC1 signaling in several tissues, including the hypothalamus. In neurons, glutamatergic activity activates extracellular signal-related kinase (ERK) along with mTORC1, consistent with our observation of P-ERK in IAA deficiency. While mTORC1 is sensitive to rapamycin (Rap), mTORC2 is not; mTORC2 at appropriate doses is sensitive to the mTOR inhibitor, Wortmannin (Wort). We studied the effect of these two mTOR complexes on the feeding responses to IAAD diets after injecting Rap or Wort into the APC. Neither affected the rejection of the IAAD diet, but intake of the second meal was increased at 40 min after a Wort injection, reviewed by [75]. This suggested that mTOR could be involved in the later responses, such as conditioned taste aversion (CTA). Furthermore, after ingesting and rejecting an IAAD diet, as described above, when the animals are offered a complete IAA diet, their intake of the better diet is robust, seen within approximately 30 min, as noted above. The mediobasal hypothalamus (MBH) contains nuclei of the arcuate and ventromedial areas and has been determined to have a central role in sensing AA abundance [77]. Moreover, L-leucine specifically, when injected into the hypothalamus, activates mTORC1, which is an effect that can be inhibited by Rap [48].

The ability to select for the growth-limiting IAA has been studied as well. Rats fed a lysine-deficient diet could select the water bottle containing lysine from several bottles, including other IAAs, or the non-nutritive sweetener, saccharin, or NaCl. When given a lysine replete diet, so that they no longer needed the lysine in their drinking water, the rats switched to the saccharin bottle [78]. Selection among IAAD diets after 2–3 days of choice showed that animals can select against the more deficient diet that contained just 0.012% less of the limiting IAA [79]. Clearly, the sensitivity to the concentrations of IAAs is acute. Flavored solutions paired with complete or IAAD diets verified that rats can develop a learned preference and learned aversion to these diets, respectively [80]. Additional reports of selection for the necessary IAA include [63,81,82]. Several species, including those of the rodent and bird order and class, respectively, can combine intake of two IAAD diets that are limiting in different IAAs [83,84]. Humans have also developed the use of complementary amino-acid-containing foods in their diets. Examples include beans and rice, used in many cultures, reviewed in [2]. A guide for achieving complimentary AA profiles, particularly for vegetarian diets, is available [85].

## 6. Conclusions

In summary, both IAAs and non-IAAs act nutritionally and in signaling roles. Protein synthesis is dependent on a full complementary array of IAAs, but in addition, homeostasis throughout the body requires dispensable AAs as well. Many aspects of AA nutrition depend on an understanding of the complex interactions among brain circuits, including both neurons and glia, and within various cell types in the periphery. In this review, we have focused on signaling within the brain. The depletion of any IAA in the plasma, and competition for transporters at the blood–brain barrier results in depletion of the limiting IAA in the brain. This can cause activation of the GCN2 pathway, which inhibits protein synthesis at the tRNA level. Signals from the APC to the lateral hypothalamus and several other brain areas are involved in activating behavioral responses.

The ability of omnivores to select a complete diet depends on the sensing of IAAD in their current diet, and the motivation to select a food containing the required complement to any IAA that is missing in the diet at hand. The development of aversion to a deficient diet involves brain areas including the amygdala and the insular cortex.

Repletion of a limiting IAA, such as leucine, activates mTORC1, while mTOR is regulated by AA transporters that include the non-IAA glutamine. The full description of how cells organize the interactions between signals for optimal metabolism and growth is not yet established. For example, increasing an IAA can prevent seizures, but in other diseases such as obesity, IAA restriction is the better option. Clearly, the knowledge of the complexities of AA interactions and effects is far from complete, and further work to understand these complex metabolic and systemic systems remains a challenge for the future.

## Figures and Tables

**Figure 1 jcm-11-00191-f001:**
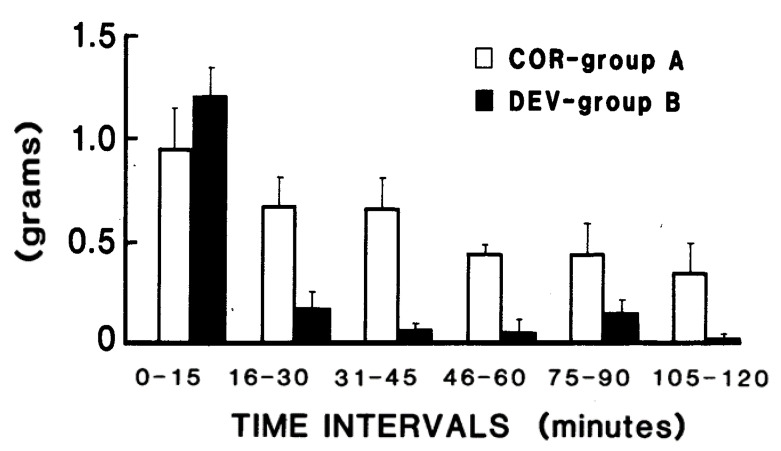
Food intake in intervals of 15 min. by rats (5/group) given corrected (COR: control) or threonine devoid (DEV) diets. Decreased food intake is seen by the DEV group in the 16–30 min period. (Taken from Gietzen et al., 1986).

**Figure 2 jcm-11-00191-f002:**
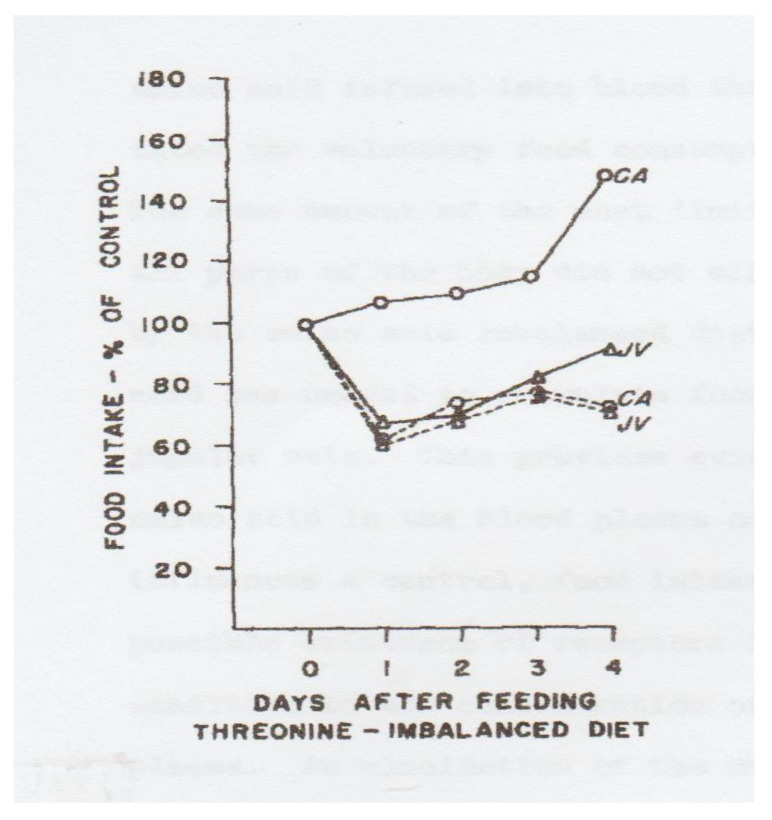
Evidence for brain recognition of IAAD diet. Results from day 1 as % of control: CA, threonine infusion: 105%. Other groups, approximately 60%. Food Intake, Percent of control on day 1 of infusion. Abbreviations: solid lines: CA, carotid artery infusions of Thr; JV, jugular infusion of Thr. Dotted lines, infusions of saline into both sites, as indicated. Image is Figure 2 from Leung and Rogers [20], with permission.

**Figure 4 jcm-11-00191-f004:**
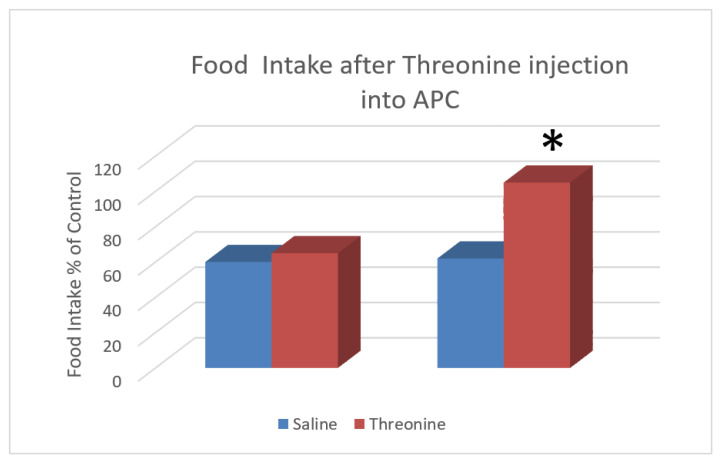
Food intake of a threonine imbalanced diet after bilateral micro-injections into the APC: Intake presented as % of each animal’s baseline intake. Injections were 2 nanomoles L-threonine into each side. The results supported the role of the APC in recognition of IAA deficiency [32]. The star * indicates significance at *p* < 0.05.

**Figure 5 jcm-11-00191-f005:**
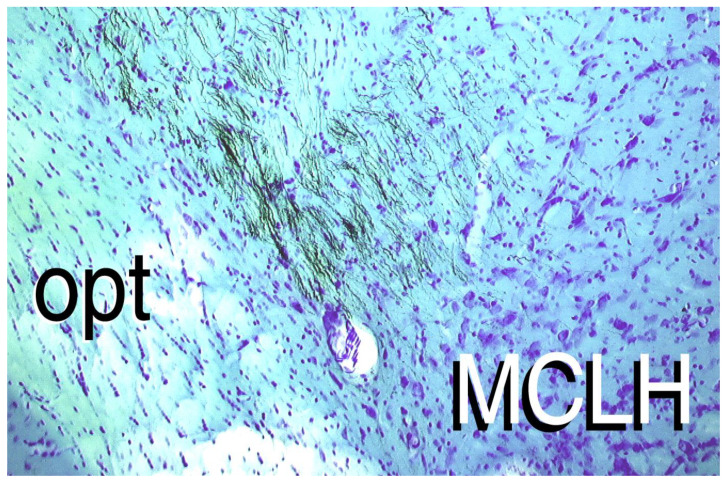
Projections from the APC to the lateral hypothalamus. Black marks show axons from the APC ending in the lateral hypothalamus. Injections of the axon marker, biocytin, into the APC make direct contact in the MCLH, the magnocellular area of the lateral hypothalamus. Figure is from Aja, S., Dissertation, University of California Davis, 1999. Opt: optic tract, seen in the white areas. Purple spots indicate cell bodies.

**Table 1 jcm-11-00191-t001:** Indispensable and Dispensable Amino Acids.

Indispensable Amino Acids	Dispensable Amino Acids
Histidine (H, His)	Alanine (A, Ala)
Isoleucine (I, Ile)	Arginine (R, Arg)
Leucine (L Leu)	Asparagine (N, Asn)
Lysine (K, Lys)	Aspartic Acid (D, Asp)
Methionine (M, Met)	Cysteine (C, Cys)
Phenylalanine (P, Phe)	Glutamic Acid (E, Glu)
Threonine (T, Thr)	Glutamine (Q, Gln)
Tryptophan, (W, Trp)	Glycine (G, Gly)
Valine (V, Val)	Proline (P, Pro)
	Serine (S, Ser)
	Tyrosine (Y, Tyr)

Abbreviations: Single letters and three-letter abbreviations assigned to each amino acid. IUPAC-IUB Joint Commission on Biochemical Nomenclature (JCBN) Nomenclature and Symbolism for Amino, Acids and Peptides.

**Table 2 jcm-11-00191-t002:** **A**: Brain areas studied. **B**: Effects of electrolytic lesions on intake of an IAAD diet, **C**: Concentrations of the limiting IAA, threonine, in brain areas as listed. Diets were based on a basal or corrected diet, each containing all IAAs or an IAA imbalanced diet, using threonine as the limiting IAA.

Brain Area	Effects of Lesions on Intake of Deficient Diet	Concentrations of Threonine
		µm/g Wet Tissue
A	B	C
**AMYGDALA**	↑ *	↑ 7% NS
**ANTERIOR CINGULATE CORTEX**	--	↓ 50%, * *p* < 0.05
**ANTERIOR PIRIFORM CORTEX**	↑	↓ 43%, * *p* < 0.05
**LATERAL HYPOTHALAMUS**	--	↑ 16% NS
**LOCUS COERULUS**	--	↓ 30% NS
**NUCLEUS OF SOLITARY TRACT**	--	↓ 43%, * *p* < 0.05%
**VENTRAL TEGMENTAL AREA**	↓	↓ 24% NS
**VENTROMEDIAL HYPOTHALAMUS**	--	↑ 2% NS
**PITUITARY**	↓	N/D
**OLFACTORY BULB**	--	N/D

Symbols used: In Column **B**: ↑ * = significant increase; arrows without * = not significant; --, no effect. In Column **C**: NS, not significant; N/D, not determined; *p* < 0.05, level of significance. In Column **B**, Behavioral Effects show altered intake of a threonine imbalanced diet introduced after 1 week of eating a threonine basal diet (see Methods for details of feeding protocol). In Column **C**, Biochemical effects: Concentration changes of the limiting IAA after 2.5 h exposure to a threonine imbalanced diet, compared to the basal control. Areas that showed no change in either measure but were not included in this table: Area Postrema, Dorsolateral Hippocampus, Lateral septum, Raphe nuclei, Thalamic taste nuclei. Data are taken from Table 1 and Table 2 in [26].

## Data Availability

Each paper cited in this review has its own data availability.

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
