# Peer review of "Brain Signaling of Indispensable Amino Acid Deficiency"

_jcm, 2021, doi:10.3390/jcm11010191_

Round 1

Reviewer 1 Report

The present review by Gietzen is a comprehensive overview of the knowledge to date regarding brain detection of dietary intake of indispensable amino acids. 

The review is well-written and in my opinion requires only minor revision. 

1) Figure 1 is blurry - is it possible to get this figure in a higher resolution?

2) Double-check figure and table legends, especially the legend of table 2 - it says that a down-arrow means "no significance", is this correct?

3) Line 246: “c-fo,s”

4) Page 8 - the author states that conditioned taste aversions develop in response to unpleasant visceral stimuli, and that IAAD diets cause conditioned taste aversions. Do IAAD diets provide such an unpleasant visceral stimulus?

5) Conclusions: The second sentence seems out of place in the context of the manuscript and could be re-written without the human part.

Author Response

Dear Reviewer:

Thank you for your kind words about my manuscript, and for your helpful comments.  In response to your suggestions, 

1) Figure 1, blurry:  I have decreased the size of the figure so that it is less blurry

2) Table 2: the legend has been edited for clarity.   The remaining legends have been reviewed and edited as necessary. 

3) line 246:  the comma in c-fos has been removed. 

4) re: negative visceral response in CTA, a citation indicating this has been added.

5) The conclusions have been re-written, and the human part has been deleted

5) Conclusions have been rewritten. 

Reviewer 2 Report

In the Review-paper entitled “Brain Signaling of Indispensable Amino Acid Deficiency” author reviewed studies describing location of the detector for IAA deficiency in the brain, with focus on recognition of IAA deficient diet in the anterior piriform cortex.

Protein is a major dietetic nutrient required for supporting animal life. Amino acids are essential for many physiological processes and IAAD diets relies on multiple sensors in both peripheral and central nervous systems.

Unfortunately, this review does not give solid overview about brain signaling of indispensable amino acids. There are several, not discussed, but essential topics for understanding the signaling pathway of IAA, e.g.:

- the role of distinct neuron populations in the central nucleus of amygdala that are involved in different/opposing feeding regulation;

- the specific role of IAA for proper neurotransmission;

- the impact of brain IAA concentrations on signaling pathway and central amino acid sensing;

- molecular marker for amino acid sensing neurons to characterize the neurocircuits engaged downstream from primary brain sensors;

- neuronal-astrocytic metabolic pathway;

- the role of IAA in glutamate-GABA homeostasis;

- how do APC interact with other AA-sensing circuits;

- the possible importance of IAA for maintaining brain function;

- methods for amino acid analysis;

- amino acid metabolism disorders, behavioral abnormalities.

Author Response

Dear Reviewer:  Thank you for your thorough review of the manuscript.  Below I list the essential topics to include, and the location of my responses:

  1. Distinct neuron populations in the Central nucleus of the Amygdala: discussed in lines 272-276.
  2. The loss of GABA input is controlled in numerous circuits.  the loss of GABAergic inhibitory neurons in IAA depletion is covered in lines 192-196.
  3. Impact of IAA loss -- loss of GABA's inhibitory signaling due to IAA depletion is also covered in answer to #2.
  4. The molecular marker for downstream circuits can be seen in Figure 5, legend on lines 211-214, and is also discussed in lines 215-225.
  5. neuro-glial pathway is noted in the conclusions, lines 312-315
  6. re: GABA- glutamate role: particularly important in the APC, mentioned in abstract, lines 13 ,14, and throughout the discussion of the mechanisms in the APC, where loss of GABA function becomes disinhibitory.
  7. maintenance of brain function:  IAA depletion can lead to seizures, lines 24-25 and conclusions
  8. Amino acid analysis: described on lines 130-132
  9. disorders and behavioral issues:  These topics have been added to lines 24-25 and 331-332

Reviewer 3 Report

Reus, December 15, 2021

I have reviewed the work of Dorothy W. Gietzen, titled: “Brain Signaling of Indispensable Amino Acid Deficiency“. This work is a review paper, This work is a review paper, whose main focus is to understand the regulatory mechanism (s) of essential amino acids at the central nervous system level. The review shows interesting and relevant findings on the role of essential amino acids. This study details historical aspects about the role of essential amino acid deficiency in animals and its implication in growth, the usefulness of using adequate protein substitutes and the regulatory role of these amino acids in the intake.

Other points reviewed are how the study of brain receptors was carried out, the anterior piriformis cortex being key to understanding the mechanism involved in the balance of essential amino acids. In addition, different studies carried out to understand the methods of control and intake of essential amino acids and the signaling mechanisms are detailed.

However, after extensive review by the author, a more interesting conclusion would be desirable. It would be convenient to point out future guidelines on the role of essential amino acids and their possible role in different pathological conditions, such as diabetes mellitus, obesity, etc.

Author Response

For the Reviewer:

Thank you for your kind review and helpful suggestions for the conclusion section.   For the conclusions section, I have added comments on the future needs for research included needed research on AA use in pathological conditions on lines 341-347.  

Round 2

Reviewer 2 Report

The review has been substantially improved and is now suitable for publication.